# Parameter Tuning of a Vapor Cycle System for a Surveillance Aircraft

**Adelia Darlene Drego [1],\*, Daniel Andersson [2] and Ingo Staack [3]**

1   Saab AB, Olof Palmes Gata 17, 5 tr, 111 22 Stockholm, Sweden
2   Modelon AB, Scheelevägen 17, 223 63 Lund, Sweden; daniel.andersson@modelon.com
3   Institute of Aircraft Design and Lightweight Structures, Technische Universität Braunschweig, Universitätspl. 2, 38106 Braunschweig, Germany; ingo.staack@tu-braunschweig.de
\*   Correspondence: adelia.drego@saabgroup.com or adelia.drego@liu.se

**Abstract:** Surveillance aircraft perform long-duration missions (>eight hours) that include detection and identification of objects on the ground, the water, or in the air. They have surveillance systems that require large amounts of cooling power (typically 10 s of kW) for long durations. For aircraft application, vapor cycle systems (VCS) are emerging as a more efficient alternative to conventional cooling systems. In this study, a two-part method was applied to a cooling system with a VCS that can be installed on a surveillance aircraft. The first part focused on a parameter tuning study set-up and demonstrated how after identifying the operating conditions, constraints, and requirements, the only cooling system parameter available for tuning was the VCS compressor speed. The second part focused on a modelling and solving strategy for the cooling system and showed how the capacity of an aircraft cooling system was impacted by tuning the VCS compressor speed (Hz) for a surveillance system heat flow rate from 10 kW to 70 kW. The results from this study can be used to design a control strategy for the compressor. In a broader perspective, the two-part method and the results analysis presented can serve as a preliminary method for aircraft VCS control optimization studies.

**Keywords:** cooling system; surveillance aircraft; vapor cycle system

## 1. Introduction

Military surveillance aircraft typically conduct missions that entail long-range detection and identification of objects over land, in the air, and at sea. These aircraft are equipped with a suite of active and passive sensors that are collectively known as an 'airborne early warning and control' (AEW&C) system. These aircraft perform long duration missions (>eight to ten hours) [1]. They are typically distinguished by the pod of electronics mounted atop the fuselage and they usually run their surveillance systems continuously on high power for several hours during a single mission. Further, the surveillance systems are temperature sensitive and must be maintained within very stringent temperature limits for safe operation. Therefore, the cooling system of a surveillance aircraft is required to have a large cooling capacity (in the 10 s of kW) and be able to provide it for long periods of time. Collection and transportation of these heat loads requires a highly efficient cooling system.

Over the last three decades, vapor cycle systems (VCS) have emerged as a more efficient alternative to conventional aircraft cooling systems such as air cycle machines (ACM). In comparison to ACM, VCS are more energy efficient. They do not require a source of high-pressure air from engine bleed air or conventional ram air for operation, for example [2–5]. The working fluid in VCS undergoes a phase change and therefore VCS have higher transfer rates than non-phase changing cooling systems making them more efficient in weight and volume [6–12]. Another benefit of VCS is that two-phase systems use the latent heat of vaporization of the working fluid for heat dissipation and rejection, unlike a single-phase system that uses sensible heat [7,13]. Further, VCS have successfully been used on operational aircraft. Successful applications of VCS on military aircraft include the

Lockheed Martin F-22 [3,14–17]. Civilian aircraft applications include Embraer Phenom 100 and Embraer Phenom 300 [18]. However, despite the benefits and real-life use of VCS for aircraft, there are other challenges for cooling systems on surveillance aircraft.

Saab AB has modified several passenger aircraft to be used for airborne surveillance. These include the indigenous Saab 340 and Saab 2000, and the Bombardier Global 6000. These aircraft have been retrofitted with AEW&C systems to conduct military surveillance missions. These surveillance systems require efficient state-of-the-art cooling systems. However, retrofitting new cooling systems aboard an existing aircraft can prove challenging. First, installation and routing of pipes can be a challenge due to limited space, blocked pathways, separation criteria, and maintenance aspects amongst other reasons. Second, the volume and weight of major cooling system components such as heat exchangers may also be limited due to limited available space and drag considerations. This impacts the cooling capacity of the system. Finally, there are a limited number of available terminal heat sinks. They are the destination of thermal energy [19] and in the case of an aircraft this would be overboard. van Heerden et al. (2022) [19] (pp. 4–6, 12) and Pal and Severson (2017) [20] (p. 800) categorized ram air, engine air streams (e.g., fan), ambient air, dissipation through airframe skin, and fuel as terminal heat sinks on an aircraft. However, the availability of terminal heat sinks can be exacerbated when retrofitting new cooling systems on existing aircraft. This is due to the challenge of routing additional coolant pipes to dump thermal energy to an available terminal heat sink. Therefore, there are several factors that limit the options of a cooling system designer in meeting the large cooling needs of surveillance systems.

Constraints due to retrofitting new systems on an existing aircraft limit the number and range of cooling system parameters that can be manipulated to meet the cooling requirements of a surveillance system. Modelling and simulation of a cooling system is a means for analyzing the impact on the cooling capacity of the system by tuning a limited number of parameters. In recent years, several studies on VCS for aircraft applications have focused on analyzing various control strategies for the system. These include either modelling and simulation or experimental set-up studies [5,12,21–24]. This paper has a different focus. The study serves as a pre-design analysis for optimizing the control of VCS components and therefore, enables the aircraft cooling system designer to conduct the following:

- Identify the design parameters of the cooling system.
- Investigate the impact on the cooling capacity of the system by manipulating the design parameters and thereby understand the performance limits of the system.
- Optimize the control strategy of the VCS and its components for static and transient operation.

### 1.1. Purpose of Paper

The first purpose of this paper is to demonstrate how operating conditions, requirements, and constraints impact the availability of aircraft cooling system parameters. Second, to demonstrate how to set up a parameter tuning study. Third, to demonstrate a computational modelling and simulation strategy for a cooling system consisting of a VCS and thereby, to use this strategy not only to find the limits of the system for various operating conditions but also to obtain a detailed performance analysis of the system components. Finally, to demonstrate how simulation results can be used to optimize the control strategy of the system components.

To meet these goals a two-part method is described in detail in this paper. The first part focuses on the parameter tuning study set-up and the second part on the modelling and solving strategy for the cooling system at hand. The results from applying this method serve as a pre-design analysis for optimizing the control strategy of the VCS and its components.

The method used to evaluate an aircraft cooling system consists of a VCS. The cooling system may be installed on a passenger aircraft to be converted to a military surveillance aircraft. The simulation approach allows for tuning of the cooling system parameters at the

component level to observe if system level requirements are met. Further, the approach demonstrates how the appropriate level of detail for each system component is chosen based on the needs of the model.

### 1.2. Outline of Paper

First, the components of the cooling system are described in Section 2. Then, the first and second part of the two-part method applied in this study are described in Sections 3 and 4, respectively. The results obtained from the system simulations are presented and discussed in Section 5. Finally, the conclusions of this study are noted in Section 6.

### 2. The Cooling System

The cooling system in this study is made up of two systems that are coupled. A liquid loop system with ethylene glycol (50%) is marked in blue and a vapor cycle system with refrigerant R134 is marked in green, respectively, in Figure 1. Ethylene glycol has been mentioned [18] or used [5,8] as a working fluid in recent studies on aircraft thermal management. Refrigerant R134 was chosen for this study because it is a commonly used refrigerants in studies [5,21–23] and real-life applications of VCS [3] in aircraft in the last 30 years. The liquid loop system is a single-phase system since in this study ethylene glycol does not change phase. The vapor cycle system with R134 changing phase is a two-phase system. The function of the components of the cooling system are described in Sections 2.1 and 2.2.

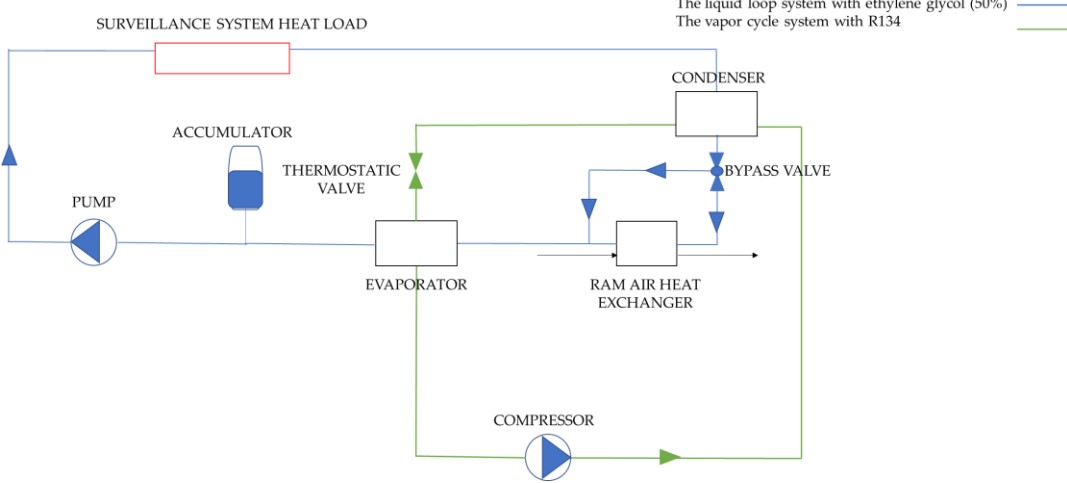

**Figure 1.** The cooling system consists of a liquid loop system coupled to a vapor cycle system.

### 2.1. The Liquid Loop System

In the single-phase system, the ethylene glycol mixture is pumped through the surveillance system where it picks up the heat load from the system. The warm ethylene glycol mixture then goes through the secondary side of a two-phase counterflow heat exchanger that acts as a condenser to the refrigerant R134. The refrigerant condenses on the primary side of the condenser while the temperature of the glycol mixture increases. As noted in Figure 1, the cooling system in this study has a single terminal heat sink, i.e., the ram air heat exchanger. This heat exchanger is a crossflow liquid air heat exchanger where the glycol on the primary side is cooled by the ram air on the secondary side. Therefore, the heat produced by the surveillance system as well as the cooling system itself, is dumped overboard via the ram heat exchanger.

If the temperature of the ethylene glycol mixture cooling the surveillance system is lower than that of the surrounding air, condensation will develop on the system electronics. To avoid condensation for the operating conditions of the aircraft, the ethylene glycol threshold temperature determining the bypass valve operation is set to 57 °C. When the

surveillance system is running on high power, the temperature of the glycol can easily surpass 57 °C and then the bypass valve stays only partially opened. Therefore, most of the ethylene glycol mixture is cooled by the ram air heat exchanger and only a fraction of the mixture bypasses the ram heat exchanger directly into the evaporator. When the surveillance system is running on low power then the glycol temperature can be well below 57 °C. The bypass valve is then (almost) fully opened and most of the mixture bypasses the ram heat exchanger directly into the evaporator.

The evaporator, like the condenser, is a two-phase counterflow heat exchanger that acts as an evaporator to the refrigerant. Therefore, the refrigerant evaporates on the primary side of the evaporator while the glycol cools down on the secondary side. The function of the pump is to pump the ethylene glycol mixture through the liquid loop at the required mass flow rate to ensure safe operation of the surveillance system. Finally, the required mass flow rate of the ethylene glycol is determined by the heat flow rate and the operating temperature range of the surveillance system. This is mathematically demonstrated in Section 4.

### 2.2. The Vapor Cycle System

The components of the two-phase loop entail the compressor, thermostatic valve, condenser, and evaporator. The condenser and evaporator are coupled to the single-phase loop. The refrigerant modelled and simulated in this study is R134. The function of the thermostatic valve is to control the temperature of the refrigerant at the evaporator outlet. To ensure the safe operation of the compressor and no liquid build-up the refrigerant must leave the evaporator as a superheated vapor. The thermostatic valve adjusts the mass flow rate of the refrigerant through the valve to ensure that the refrigerant leaves the evaporator and enters the compressor as a superheated vapor. The compressor is a variable speed compressor with constant volume displacement that compresses the refrigerant and pumps it towards the condenser.

## 3. Method Part 1: Parameter Tuning Study Set-Up

The first part of the method for this study is the parameter tuning study set-up that precedes the modelling and solving strategy for the cooling system. The workflow for the tuning study set-up is pictorially shown in Figure 2. First, the operating conditions of the aircraft and surveillance system are identified and quantified. They are described in detail below in Section 3.1. Similarly, the functional requirements and constraints on the cooling system are identified. They are described in Section 3.2. The operating conditions and requirements are used to identify the tunable characteristics of the cooling system, and this is described in Section 3.3. Finally, this leads to the cooling system simulation set-up described in Section 3.4.

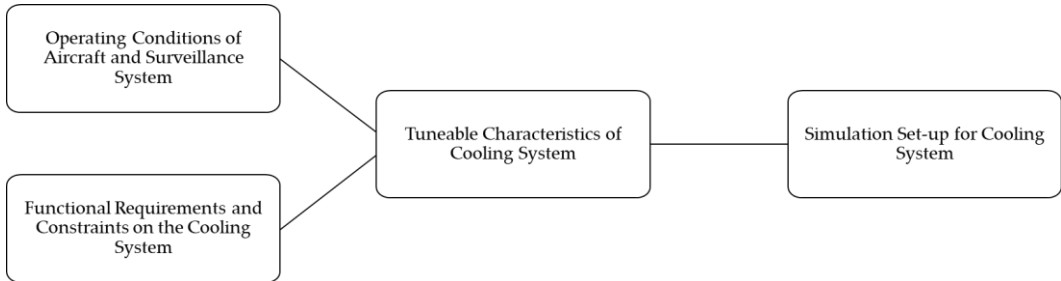

**Figure 2.** Method workflow for parameter tuning study set-up for cooling system.

### 3.1. The Operating Conditions of the Aircraft and Surveillance System

The detection and identification characteristics of a surveillance system determine the optimal operating conditions of the aircraft in terms of altitude and Mach number (M). The cooling system model in this study is simulated for two operating points of the surveillance aircraft. The operating conditions of the aircraft are defined in terms of altitude and Mach

number and they determine the ram air conditions at the ram air heat exchanger inlet for each operating point. The first operating point is at an altitude of 6.5 km and a flight Mach number of 0.4. The second operating point is at 11 km and a flight Mach number of 0.55. The ambient temperature, $T_{AMB}$ at each operating point is calculated assuming International Standard Atmosphere +15 °C (ISA + 15). The ram air temperature, $T_{RAM}$, at each operating point is calculated using Equation (1). Operating point data are summarized in Table 1.

$$T_{RAM} = T_{AMB}\left(1 + 0.2M^2\right) \tag{1}$$

**Table 1.** Operating conditions for aircraft.

| Operating Point | Altitude (km) | Mach Number | $T_{AMB}$ (°C) | $T_{RAM}$ (°C) |
|:---:|:---:|:---:|:---:|:---:|
| 1 | 6.5 | 0.4 | −12 | −3 |
| 2 | 11 | 0.55 | −42 | −28 |

To find the limits of the cooling system, the surveillance system electrical power consumption is assumed to be between 10 kW and 70 kW at both operating points. Assuming the surveillance system has zero efficiency, then the heat flow rate of the system is equal to the electrical power it draws. In this study, the surveillance system heat flow rate is tuned from 10 kW to 70 kW in intervals of 10 kW at each operating point.

*3.2. Functional Requirements and Constraints on the Cooling System*

The distinction between functional requirements and constraints is adopted from [25] (p. 103). The minimum set of independent requirements that wholly distinguish the functional needs of the product or system are functional requirements. On the other hand, constraints are non-functional requirements and can be classed as input or system constraints. Input constraints apply to all concepts while system constraints are specific to a concept [25]. For example, the cooling system in this study has a single terminal heat sink, i.e., ram air through the ram air heat exchanger. This system constraint results in the cooling system dumping the heat load overboard through only one terminal heat sink. A functional requirement and three input constraints imposed on the cooling system to be retrofitted into the surveillance aircraft are described in detail below. However, to keep track of the many requirements and constraints on a system in a large aircraft project a commercial tool such as IBM Doors [26] can be used.

- Functional Requirement 1: The cooling system must collect the heat load produced by the surveillance system and dispose it overboard.
- Input Constraint 1: The first and foremost input constraint that the cooling system must fulfil is to ensure that the surveillance system is maintained within its safe operating temperature range. This means that the temperature of the ethylene glycol mixture at the inlet to the surveillance system must never exceed 30 °C and its temperature at the outlet of the system must never exceed 45 °C. The mass flow rate of the ethylene glycol mixture and the R134 refrigerant determine if this input constraint is met at each heat flow rate setting of the surveillance system from 10 kW to 70 kW. The mass flow rate of R134 and ethylene glycol is determined by the compressor speed and pump speed, respectively. To partly fulfil this constraint, ethylene glycol must be pumped at a specific mass flow rate for a given heat flow rate of the surveillance system, $\dot{Q}_{surveillance\_sys}$. The mass flow rate of ethylene glycol, $\dot{m}_{glycol}$ is calculated using Equation (2):

$$\dot{m}_{glycol} = \frac{\dot{Q}_{surveillance\_sys}}{\Delta T_{surveillance\_sys} c_{glycol}} \tag{2}$$

where $\Delta T_{surveillance\_sys}$ is the temperature difference across the inlet and outlet of the surveillance system and $c_{glycol}$ is the specific heat capacity of ethylene glycol (50%).

For the complete cooling power range of the surveillance system in increments of 10 kW, the mass flow rate of the ethylene glycol is shown in Table 2. Here, $\Delta T_{surveillance\_sys} = 45\,°C - 30\,°C = 15\,°C$ and $c_{glycol} = 3.5 \frac{kJ}{kg} \cdot K$. Note that a constant value for $c_{glycol}$ is used to calculate $\dot{m}_{glycol}$. However, for the model simulations, $c_{glycol}$ is temperature dependent. This results in minor variations in expected and actual $\Delta T_{surveillance\_sys}$.

**Table 2.** Corresponding mass flow rates of ethylene-glycol mixture, $\dot{m}_{glycol}$ for heat flow rates of the surveillance system, $\dot{Q}_{surveillance\_sys}$.

| $\dot{Q}_{surveillance\_sys}$ (kW) | $\dot{m}_{glycol}$ (kg/s) |
|---|---|
| 10 | 0.19 |
| 20 | 0.38 |
| 30 | 0.57 |
| 40 | 0.76 |
| 50 | 0.95 |
| 60 | 1.14 |
| 70 | 1.33 |

- Input Constraint 2: The second input constraint is set by the conditions at the inlet duct to the ram air heat exchanger. The duct is ram-pressure driven with no forced suction. Therefore, the total pressure in the stream tube, $p_{t\infty}$, is assumed to be the same as the total pressure in the throat area of the duct, $p_{t\_Tr}$. This results in lip losses having to be minimised. Therefore, the cross-sectional capture area at the inlet, $A_C$ must be greater than the cross-sectional capture area in the stream tube, $A_\infty$. Areas and pressures in the stream tube and inlet are indicated in Figure 3. For a cross-sectional area of the throat, $A_{Tr}$ of 0.04 m$^2$ and assuming that

$$A_C = 1.1 A_{Tr} \tag{3}$$

then, $A_C$ is 0.44 m$^2$.

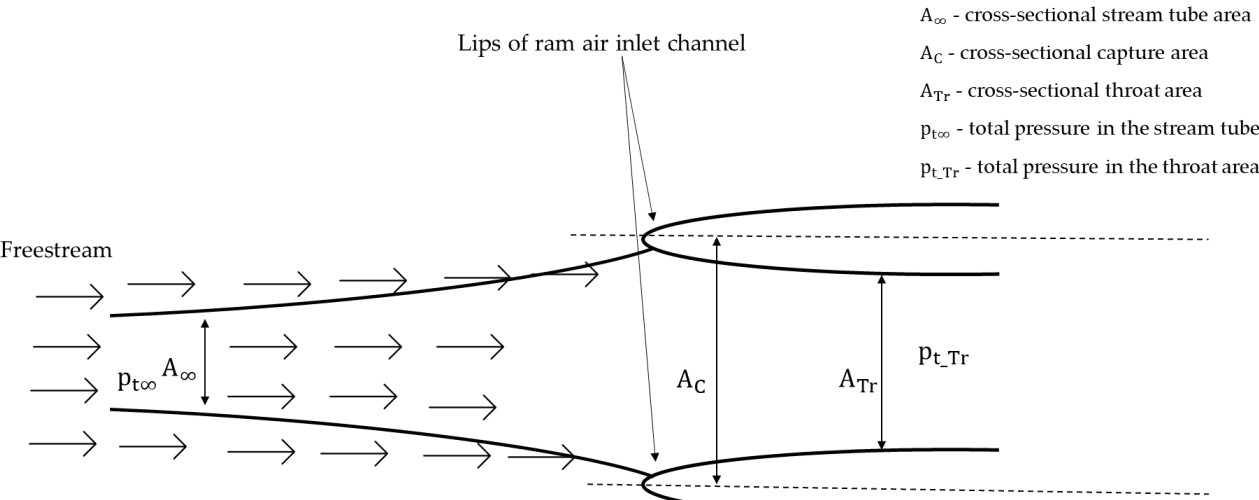

A$_\infty$ - cross-sectional stream tube area
A$_C$ - cross-sectional capture area
A$_{Tr}$ - cross-sectional throat area
p$_{t\infty}$ - total pressure in the stream tube
p$_{t\_Tr}$ - total pressure in the throat area

**Figure 3.** The location of the stream tube, capture, and throat cross-sectional areas for a ram air inlet are indicated. The total pressure locations in the stream tube and throat are also indicated.

To minimize lip losses, the mass flow capture ratio (MFCR) should be less than one. The MFCR is defined as follows:

$$\text{MFCR} = \frac{A_\infty}{A_C} = \frac{\dot{m}_{ram}}{A_C \rho_\infty V_\infty} \tag{4}$$

$\rho_\infty$, $V_\infty$ are the density and velocity of air in the stream tube.

Assuming ISA + 15 at both operating points, then at

○ Operating point 1: ISA+15, $\rho_\infty$ = 0.6 kg/m$^3$ and $V_\infty$ = 130 m/s

○ Operating point 2: ISA+15, $\rho_\infty$ = 0.37 kg/m$^3$ and $V_\infty$ = 169 m/s

For an inlet ram air mass flow rate, $\dot{m}_{ram}$ of 2 kg/s and 1.4 kg/s at operating point 1 and 2, respectively and using Equation (4), yields an MFCR = 0.6 at operating point 1 and an MFCR = 0.5 at operating point 2.

Therefore, to minimize lip losses and maintain an MFCR < 1, $\dot{m}_{ram}$ was set to 2 kg/s at operating point 1 and 1.4 kg/s at operating point 2 for all simulations in this study.

- Input Constraint 3: Retrofitting an existing aircraft with an additional cooling system leads to a very limited available volume for the components. The total available volume for the condenser and evaporator was limited to 0.15 m$^3$. Using commercial off the shelf (COTS) options to fulfil this constraint, the dimensions of the condenser and evaporator are given in Table 3. The dimensions of the ram air heat exchanger are also based on a COTS option and are shown in Table 3.

**Table 3.** Dimensions of the heat exchangers of the cooling system.

| Heat Exchanger | Height (m) | Width (m) | Length (m) | Volume (m$^3$) |
| --- | --- | --- | --- | --- |
| Condenser | 0.3 | 0.3 | 0.8 | 0.072 |
| Evaporator | 0.3 | 0.3 | 0.8 | 0.072 |
| Ram Air Heat Exchanger | 0.35 | 0.35 | 0.1 | 0.012 |

*3.3. Tuning of Cooling System Parameters*

When the input constraints are defined, then the question that crops up for a cooling system designer is 'Which parameters of the system are available for tuning?'. The ram air mass flow rate is limited by the second input constraint. For the cooling system at hand, the pump speed is predetermined by the first input constraint. The heat transfer areas of all three heat exchangers are limited by the third input constraint. The compressor of the vapor cycle system has a constant volume displacement. Change of refrigerant in the vapor cycle system might require a different-sized condenser and evaporator depending on refrigerant properties. Therefore, the only available cooling system parameter for tuning is the compressor speed (Hz) and this is pictorially shown in Figure 4. Therefore, for tuning of surveillance system heat flow rate, $\dot{Q}_{surveillance\_sys}$, compressor speed (Hz) is tuned to observe the impact on the cooling capacity (kW) of the system.

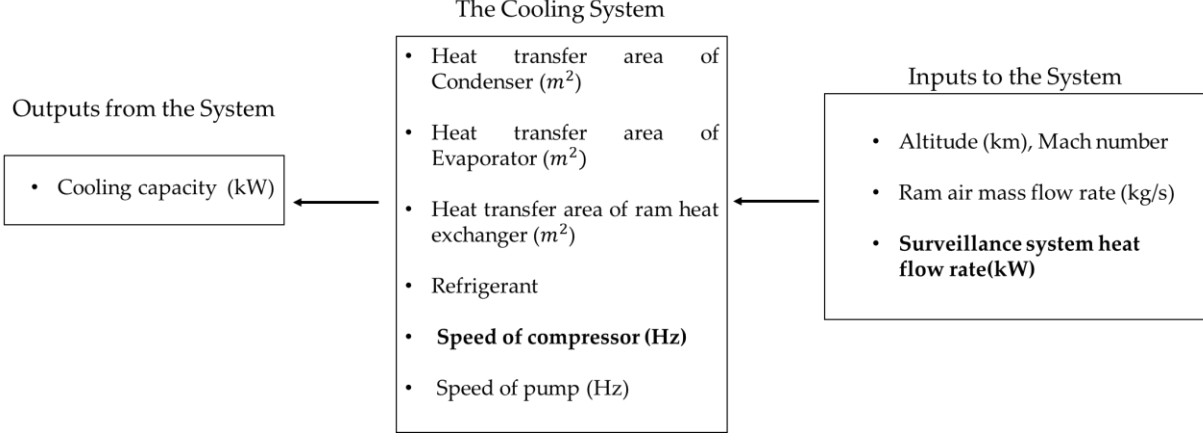

**Figure 4.** A summary of the input parameters, output parameter, and cooling system parameters. The embolden parameters are tuned to observe the impact on the output parameter, i.e., cooling capacity (kW).

*3.4. Simulation Set-Up for Cooling System*

Compressor speed (Hz) drove the simulation set-up for this study, and this is pictorially depicted in Figure 5. $\dot{Q}_{surveillance\_sys}$ at operating points 1 and 2 are tuned from 10 kW to 70 kW in increments of 10 kW. At each $\dot{Q}_{surveillance\_sys}$, the compressor speed (Hz) is tuned from 'compressor speed 1' to 'compressor speed n' to investigate if the first input constraint defined in Section 4 is met. At each compressor speed from 1 to n, a dynamic simulation was run until the solution reached steady state. Note that the value of compressor speed 1 and compressor speed n differed for each $\dot{Q}_{surveillance\_sys}$. In each simulation, the inlet and outlet temperature of the ethylene glycol across the surveillance system was monitored to ascertain if input constraint 1 was fulfilled.

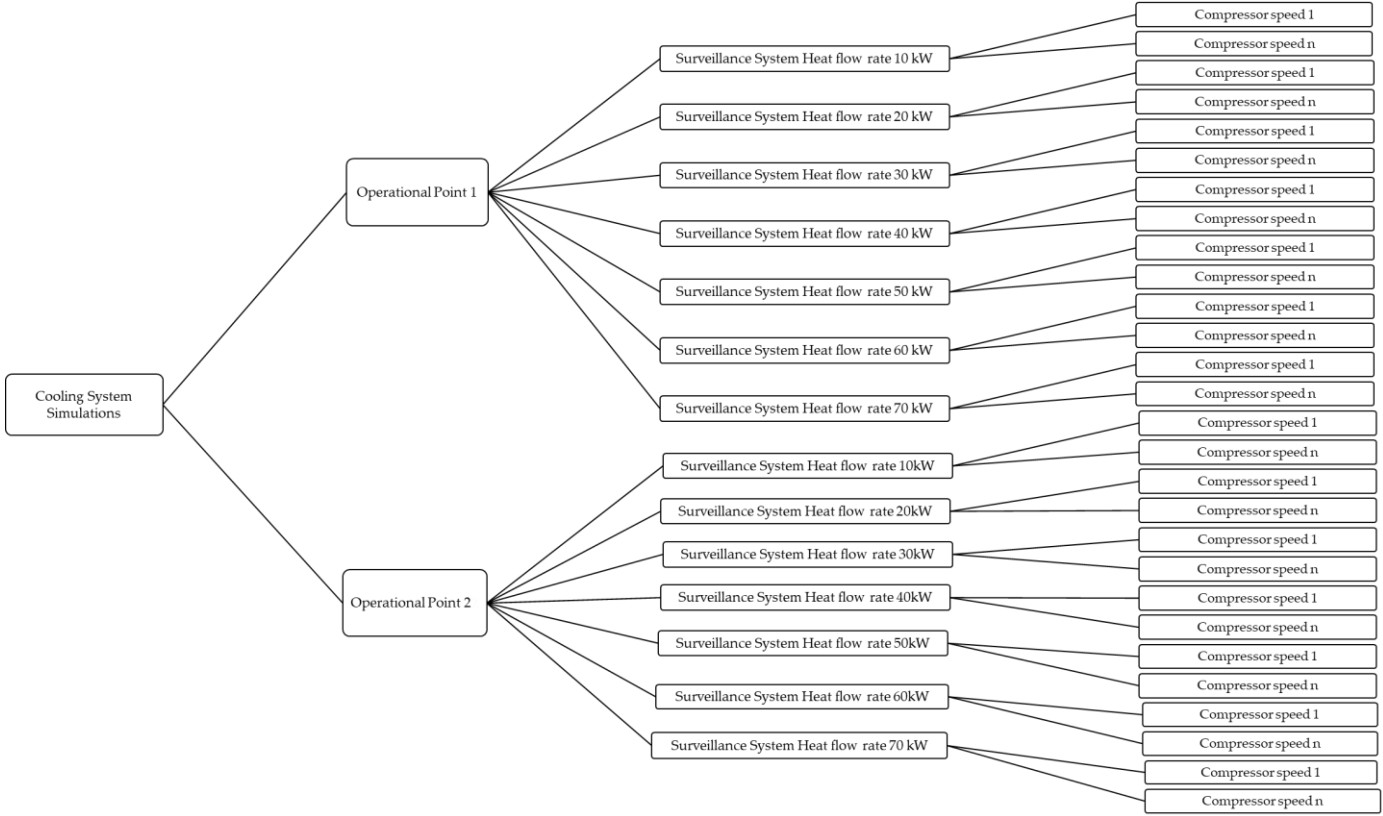

**Figure 5.** The set-up of the simulation of the cooling system.

## 4. Method Part 2: Modelling and Solving Strategy for the Cooling System

The second part of the method for this study entails the modelling and solving strategy for the cooling system. The cooling system is modelled and simulated in Modelon Impact. Modelon Impact is a systems design environment that supports system-level modelling, simulation, optimization, and analysis with Modelica-language based model libraires. All aspects of the cooling system model and how the model is solved are described in this section. This section is divided into three sub-sections, namely, 'Physical Aspects of the Cooling System Model', 'Cyber Aspects of the Cooling System Model', and 'Solving the Model in Modelon Impact' described in Sections 4.1–4.3, respectively.

*4.1. Physical Aspects of the Cooling System Model*

The physical aspects of the cooling system model entail the system hardware and the operating conditions of the aircraft and the surveillance system. Components from the Liquid Cooling library and the Vapor Cycle library in Modelon Impact are employed to model the components of the liquid loop system and the vapor cycle system, respectively.

$\dot{Q}_{surveillance\_sys}$ is also represented in the model as part of the liquid loop system. The cooling system model created in Modelon Impact is shown in Figure 6.

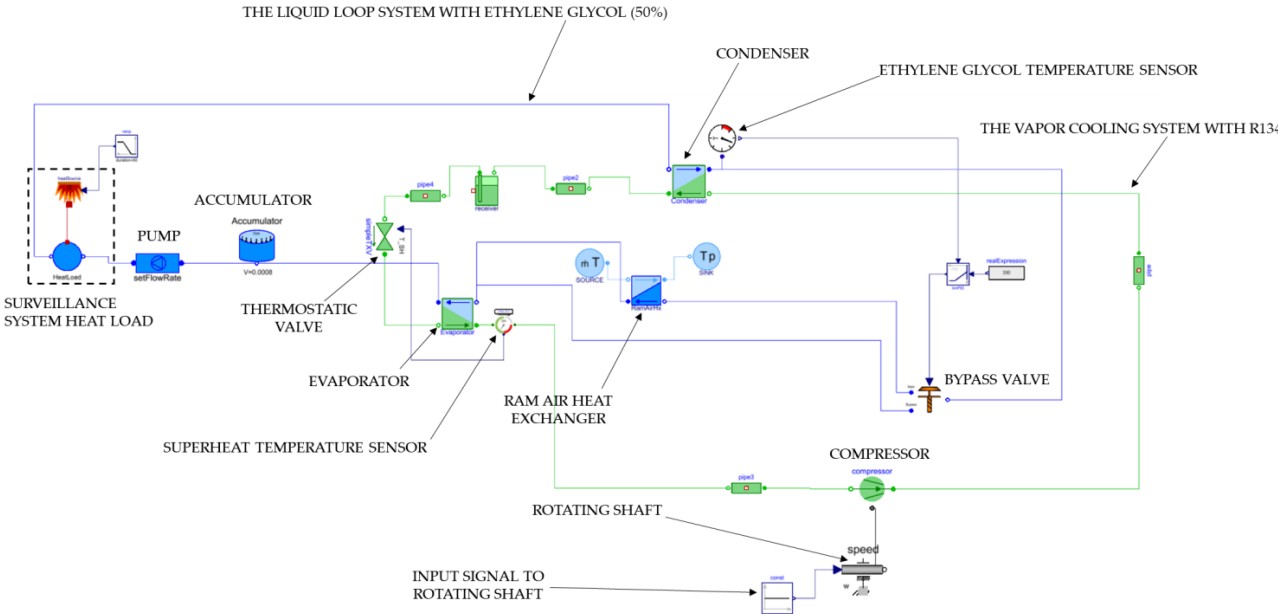

**Figure 6.** Model of the cooling system in Modelon Impact.

### 4.1.1. Surveillance System Heat Flow Rate, $\dot{Q}_{surveillance\_sys}$

$\dot{Q}_{surveillance\_sys}$ is represented using a two-port (inlet and outlet for the liquid) ideally mixed volume component from the Liquid Cooling library with a time-invariant volume. This component also has a heat port to add an external heat load and the temperature in the heat port is equal to the fluid temperature of the mixed volume. In the model, the heat port is connected to a heat source with a user-prescribed heat flow rate in W.

### 4.1.2. Ram Air Heat Exchanger

The ram air heat exchanger is represented using a liquid-gas epsilon-NTU (number of transfer units) heat exchanger component from the Liquid Cooling library. The liquid in this case is ethylene glycol on the primary side of the heat exchanger and ram air on the secondary side. Ram air is represented using fast dry air from the Modelon Media library. To be conservative, dry air is chosen over humid air since humidity can enhance the performance of the heat exchanger.

For the epsilon-NTU method employed, the heat exchanger effectiveness, $\epsilon$, is a function of NTU and the ratio of heat capacity flows of the two fluids as shown in Equation (5):

$$\epsilon = f\left(NTU, \ \frac{\dot{m}_{liquid} \ c_{liquid}}{\dot{m}_{gas} \ c_{gas}}\right) \tag{5}$$

where $\dot{m}_{liquid}$, $\dot{m}_{gas}$ are the mass flow rates of the liquid and gas, respectively and $c_{liquid}$, $c_{gas}$ are the specific heat capacities of the liquid and gas, respectively. NTU is defined as

$$NTU = \frac{UA_w}{c_{min}} \tag{6}$$

where U is the overall heat transfer coefficient and $A_w$ is the total wall heat transfer area.

Note that different functions exist for $\epsilon$ for different flow arrangements [27]. The overall heat transfer coefficient of the heat exchanger depends on heat transfer coefficients

at both wall–fluid boundaries and thermal conductivity through the wall. It is obtained by summarising thermal resistances, TR such that

$$\text{Overall TR} = \frac{1}{\text{UA}_\text{w}} \qquad (7)$$

and

$$\text{TR} = \frac{1}{\frac{1}{\alpha_\text{prim}\text{A}_\text{prim}} + \frac{1}{\alpha_\text{sec}\text{A}_\text{sec}} + \frac{t_\text{w}}{\lambda_\text{w}\text{A}_\text{w}}} \qquad (8)$$

where,

$\alpha_\text{prim}$, $\alpha_\text{sec}$ are the heat transfer coefficients for the wall-fluid boundary layer for the primary and secondary side of the heat exchanger, respectively,

$\text{A}_\text{prim}$, $\text{A}_\text{sec}$ are the heat transfer surface areas on the primary and secondary sides of the heat exchanger, respectively,

$t_\text{w}$ is the wall thickness and

$\lambda_\text{w}$ is the thermal conductivity of the wall material.

The total wall heat transfer area, $\text{A}_\text{w}$ is calculated to be 19.6 m$^2$ using the heat transfer area model by [27] (pp. 210–212) given by

$$\text{A}_\text{w} = 1600 * l * b * w \qquad (9)$$

where l, b, and w are the length, breadth, and width, respectively of the heat exchanger.

### 4.1.3. Ambient Conditions at the Aircraft Operating Points

The Mach number and the altitude for aircraft operating points 1 and 2 are used to calculate the ram air temperature at each of the points (see Table 1). The ram air temperature is represented in the model in the gas flow source component connected to the inlet of the secondary side of the ram air heat exchanger. The temperature and mass flow rate of the ram air are prescribed in this component when simulated for each aircraft operating point.

### 4.1.4. Pump

The pump is represented in the model with a component that sets the liquid flow rate based on a user-prescribed value. The available centrifugal pump component from the Liquid Cooling library was not used in this model because of a lack of data for pump characteristics in terms of flow rate and speed. The flow rate of the ethylene glycol mixture is set by each power setting of the surveillance system as demonstrated in Section 3.2 and then summarized in Table 2. The one drawback with using this liquid flow rate component is that pump losses and therefore the heat load from the pump is not accounted for in this model.

### 4.1.5. Bypass Valve

The bypass valve is represented using a thermostatic three-port valve component from the Liquid Cooling library that uses an external controller. The inlet flow is split into two branches by two valves. The main valve opening is controlled directly by the input control signal and the bypass valve opens as the main valve closes.

### 4.1.6. Accumulator

The accumulator is represented using an expansion volume component from the Liquid Cooling library that contains an incompressible liquid phase that is connected to the port and a trapped air mass. Pressure is determined by the gas phase depending on its compression by the liquid.

### 4.1.7. Condenser and Evaporator

The condenser and evaporator are modelled using a counter-flow heat exchanger model from the Vapor Cycle library for a two-phase fluid on the primary side and a liquid

without phase change on the secondary side. Both primary and secondary fluid channels are discretized using the finite volume method. The model assumes homogeneous two-phase flow (no slip between phases) and homogeneous flow distribution in parallel flow channels. The heat transfer area on both the primary and secondary side are equal and is calculated to be 115 m$^2$ using Equation (9).

This heat exchanger component has a flow resistance model, correlating mass flow rate with pressure drop. The actual correlation is replaceable and can be chosen by the user, either based on empirical design correlations, taking geometry and fluid transport properties into account or a correlation based on a user-specified nominal operating point. This correlation provides the specific flow rate under specified nominal pressure drop and inlet fluid density. Deviations from it for other pressure losses or density are in accordance with the Darcy–Weisbach equation.

### 4.1.8. Compressor

The compressor is modelled using a fixed (volume) displacement compressor model from the Vapor Cycle library. The volumetric displacement of the compressor in this study is $2 \times 10^{-5}$ m$^3$. The volumetric, isentropic, and mechanical efficiencies of the compressor are based on empirical data for compressor speeds (rad/s) and pressure ratios. These efficiencies are independent of the relative volume displacement as shown in Equations (10)–(12). Therefore, the mass flow rate varies proportionally to the relative volume displacement:

$$\eta_{volume} = \frac{\dot{m}_{refrigerant}}{\rho \cdot V_d \cdot \omega/2\pi} \tag{10}$$

$$\eta_{isentropic} = \frac{h_{isentropic} - h_{suction}}{h_{discharge} - h_{suction}} \tag{11}$$

$$\eta_{mech} = \frac{P_{extracted}}{P_{shaft}} = \frac{h_{isentropic} - h_{suction}}{P_{shaft}} \tag{12}$$

where

$\eta_{volume}$ is volumetric efficiency

$\dot{m}_{refrigerant}$ is the mass flow rate of refrigerant

$\rho$ is the density of the refrigerant

$V_d$ is volume displacement of the compressor

$\omega$ is compressor speed (rad/s)

$\eta_{isentropic}$ is the isentropic efficiency of the compressor

$h_{isentropic}$ is isentropic enthalpy of the compressor

$h_{suction}$ is specific enthalpy at the suction side

$h_{discharge}$ is specific enthalpy at the discharge side

$\eta_{mech}$ is the mechanical efficiency of the compressor

$P_{shaft}$ is the compressor shaft power

The boundary conditions of the compressor are the $h_{suction}$, $\omega$, inlet pressure, $p_{suction}$, and outlet pressure, $p_{discharge}$. The computed properties are $\dot{m}_{refrigerant}$, $h_{discharge}$ (from which temperature can also be computed), and shaft torque, T (from which $P_{shaft}$ can also be computed, knowing the $\omega$). The order of computations are as follows:

- $\eta_{volume}$ is obtained from empirical data for specified values of pressure ratio and $\omega$.

  $\dot{m}_{refrigerant}$ is computed according to Equation (10). $\rho$ is computed from the $p_{suction}$ and specific enthalpy.

- $\eta_{isentropic}$ relates the compression process to an isentropic compression, according to Equation (11). The specific enthalpy after isentropic compression ($h_{isentropic}$) from $p_{suction}$ to $p_{discharge}$ starting with $h_{suction}$ is obtained from the refrigerant property model. $\eta_{isentropic}$ is obtained from empirical data for specified values of $p_{suction}$, $p_{discharge}$, and $\omega$. Then, $h_{discharge}$ can be obtained using Equation (11).

- Knowing $\dot{m}$, $h_{suction}$, $h_{discharge}$, the required power, P to compress the gas can be obtained using the energy balance of the compressor given by

$$P = \dot{m}_{refrigerant} \left( h_{discharge} - h_{suction} \right) \tag{13}$$

- The real power consumption of a compressor is typically slightly higher than the value computed using Equation (12), and $\eta_{mech}$ obtained from empirical data is used to characterize it. The losses captured in $\eta_{mech}$ are the power provided to the compressor via the rotational shaft that does not reach the compressed gas due to internal friction and heat transfer from the gas to the solid parts of the compressor. Ultimately, it is transferred as heat to the surroundings. With look up tables for $\eta_{mech}$ and already computed $\dot{m}_{refrigerant}$ and $h_{discharge}$, the compressor power consumption is computed using Equation (12) and the shaft torque, T given by

$$T = \frac{P_{shaft}}{\omega} \tag{14}$$

### 4.1.9. Thermostatic Valve

The thermostatic valve is represented using a thermostatic expansion valve model from the Vapor Cycle library. This model follows the IEC 534/ISA S.75 standards for valve sizing that are valid for compressible fluids without phase change, including choked conditions for a variable flow coefficient, Kv [28]. The complete mass of the refrigerant R134 must leave the evaporator as a superheated vapor. If the refrigerant leaving the evaporator and entering the compressor did so as a saturated liquid–vapor mixture, it would lead to erosion of the compressor due to wet compression. Therefore, this valve component consists of a limited proportional integral (PI) controller that sets the flow coefficient, Kv, of the valve based on the measured superheating of the refrigerant at the evaporator outlet.

### 4.2. Cyber Aspects of the Cooling System Model

The cyber aspects of the cooling system model entail the control strategy for the bypass valve, thermostatic valve, and compressor.

### 4.2.1. Control Strategy for the Compressor Speed

As established in Section 3.4 and in Figure 5, the compressor speed (Hz) is the only parameter that is tuned in the model for a $\dot{Q}_{surveillance\_sys}$ range of 10 kW to 70 kW in intervals of 10 kW. The flange of the compressor component in the model is driven by a rotating shaft component based on speed (in Hz) where the speed is defined by an input signal to the component. The input signal is fed via a constant real input signal component that is manually updated in the model to 'tune' the compressor speed. The rotating shaft component and real input signal component are indicated in Figure 6.

### 4.2.2. Control Strategy for the Thermostatic Valve

The PI controller ensures the complete mass of the refrigerant leaving the evaporator is a superheated vapor. It does so by setting the flow coefficient Kv of the valve. The flow coefficient is adjusted based on the measured superheating of the refrigerant at the evaporator outlet. The measured superheat temperature of the refrigerant at the evaporator outlet is an input signal to the thermostatic valve component. This temperature measurement is indicated in Figure 6. Based on the input temperature, a user-defined set point superheat temperature ensures that the refrigerant is at or above its saturated vapor temperature (for a given pressure). The set point superheat temperature is 5 °C in this study.

### 4.2.3. Control Strategy for the Bypass Valve

The bypass valve is controlled using an external limited PI controller component. The setpoint temperature for the controller is 57 °C (see Section 2). The ethylene glycol temperature measured at the condenser outlet is the input signal to the PI controller. This temperature measurement is indicated in Figure 6. The opening and therefore mass flow rate of the ethylene glycol through the main and bypass branch are controlled based on the input signal value.

### 4.3. Solving the Model in Modelon Impact

All Modelica-based models consist of several differential and algebraic equations, forming a differential-algebraic system of equations (DAE). A brief overview of how Modelon Impact solves a DAE is described below.

- First the dynamic state variables are identified. The derivates of these can be solved from the DAE, and their time-dependent solution is obtained using numerical integration.
- The remaining equations of the DAE for a model are sorted such that for known values at every given time-step of parameters (i.e., constant values), boundary conditions (i.e., user-defined inputs) and dynamic state variables, all other model variables can be either explicitly computed with algebraic equations or obtained by solving linear or non-linear algebraic systems of equations.

The numerical integration of dynamic state variables is performed using an ordinary differential equation (ODE) solver. In this study, CVode is used which is a backward differentiation formula. However, the Radau method (a Runge–Kutta method) and explicit Euler methods are also available in Modelon Impact. The system of equations that defines the derivatives of dynamic states are solved, as needed by the ODE solver. If non-linear systems of equations need to be solved to obtain state derivatives, then the Newton–Raphson method (for systems of equations) or Brents method (for single equations) is used instead.

For the user-defined output interval, the values of all model variables are computed and saved to file. In this step the values of dynamic state variables are obtained by linear interpolation from the latest time-steps with solved values, and all variables are computed from the algebraic model equations. Again, Brents or the Newton–Raphson methods are used to solve any non-linear systems of equations in this step.

## 5. Results and Discussion

For the simulation set-up described in Section 3.4, the results from tuning the compressor speed (Hz) are shown in Section 5.1 through Section 5.4. Section 5.1 shows and discusses results for the overall performance of the cooling system in terms of meeting input constraint 1 imposed on the system. Sections 5.2 and 5.3 show and discuss results for the performance of the heat exchangers of the cooling system at operating points 1 and 2. Finally, Section 5.4 shows and discusses the pressure–enthalpy diagrams for the VCS at operating points 1 and 2, respectively. Therefore, the overall performance of the cooling system and the performance of components of the system are shown for the tuning of the compressor speed.

The results in Section 5.1 through Section 5.3 are presented using parallel coordinate plots. A parallel coordinate plot is used when data needs to be compared across multiple variables. The use of these plots for visualization of large data sets is described in detail in [29]. A recent example of using this type of plot in aircraft-related research can be found in [30] (p. 15). In the study at hand, at each compressor speed for each heat flow rate setting (kW), inlet and outlet temperatures, and heat flow rates were recorded for the cooling system components. To have a neat visual representation of this large multi-variable data set, parallel coordinate plots were used. These plots can be found in Section 5.1 through Section 5.3.

Results in recent studies on aircraft VCS [5,12,21–24] are presented for control strategies designed for the system and/or components for dynamic simulations of complete aircraft missions. However, in this study the results represent steady-state solutions obtained at operating points 1 and 2. These steady-state results can serve as a preliminary analysis to design a control strategy for the compressor, specific to the surveillance aircraft operating points. In a slightly broader perspective, the results analysis presented in this section can serve as a preliminary analysis method for aircraft VCS control strategies.

### 5.1. Limits of the Cooling System

Figures 7 and 8 clearly demonstrate the limits of the cooling system in meeting input constraint 1 for operating points 1 and 2, respectively. The third and fourth axis of both plots represent the temperature of ethylene glycol at the inlet and outlet to the surveillance system, respectively. The fifth axis in both plots indicates whether input constraint 1 has been violated for a given $\dot{Q}_{surveillance\_sys}$ and compressor speed. The compressor speed was increased from a starting value of 50 Hz in increments of 50 Hz. Starting at operating point 1 at 10 kW of $\dot{Q}_{surveillance\_sys}$, compressor speed 1 was 50 Hz and compressor speed n was 150 Hz, where n is 3. However, a solution could not be computed at 50 Hz. Therefore, the compressor limit for the cooling system was found to be 100 Hz. This can be noted in Figure 7 at 10 kW with only plots for compressor speed at 100 Hz and 150 Hz shown. This result is also noted for operating point 2 in Figure 8. At operating point 1 at 20 kW, from Figure 7 it can be noted that compressor speed 1 is 100 Hz, compressor speed 2 is 150 Hz, and compressor speed 3 is 200 Hz. However, at 100 Hz, input constraint 1 is not met. Therefore, the lowest speed that the compressor can run at for operating point 1 at 20 kW is 150 Hz. This is noted for operating point 2 as well at 20 kW. Therefore, the lowest speed of the compressor for each $\dot{Q}_{surveillance\_sys}$ can be noted from Figures 7 and 8. Similarly, the upper limits of the cooling system can also be noted from these two figures. At operating point 1, at 60 kW and 70 kW when tuning the compressor speed from 100 Hz to 700 Hz, input constraint 1 could not be met. This indicates that at operating point 1, the cooling system can cool a maximum of 50 kW of $\dot{Q}_{surveillance\_sys}$. Similarly, at operating point 2, the cooling system can cool a maximum of 60 kW of $\dot{Q}_{surveillance\_sys}$ for a minimum compressor speed of 550 Hz. However, at both operating points the maximum cooling capacity should be reduced by 1 kW. This is to account for heat due to pump losses that is not included in the model and in this case is assumed to be 1 kW. Therefore, the limits of the cooling system are determined from the parameter results shown in Figures 7 and 8.

### 5.2. Performance of the Condenser and the Evaporator

Figures 9 and 10 display the performance parameters for the condenser and evaporator, respectively. For each $\dot{Q}_{surveillance\_sys}$, the results displayed in these two figures are at the lowest compressor speed that does not violate input constraint 1. The compressor speed is obtained from Figures 7 and 8. The performance of the condenser is very similar at operating 1 and operating point 2 for $\dot{Q}_{surveillance\_sys}$ from 10 kW to 50 kW. This is true for the evaporator as well. In Figure 9, the fourth axis represents the heat flow rate from R134 as it condenses from a gaseous to a liquid state in the condenser. On the other hand, the fourth axis in Figure 10 represents the heat flow rate to R134 as it boils in the evaporator. From the two figures, it can be noted that the heat flow rate from R134 to the condenser is greater than the heat flow rate from the evaporator to R134 for the same $\dot{Q}_{surveillance\_sys}$. as would be expected. R134 in the condenser has a higher heat load than that of the glycol mixture in the evaporator that is then transferred to R134. The fifth and sixth axes in Figure 9 represent the temperature of R134 on the primary side of the inlet and outlet of the condenser, respectively. It can be noted that the inlet temperature of R134 for operating point 2 in Figure 9 increases by 24.3 °C for increasing $\dot{Q}_{surveillance\_sys}$. This

would be expected since for increasing $\dot{Q}_{surveillance\_sys}$, the compressor speed increases and therefore the temperature of R134 at the condenser inlet increases.

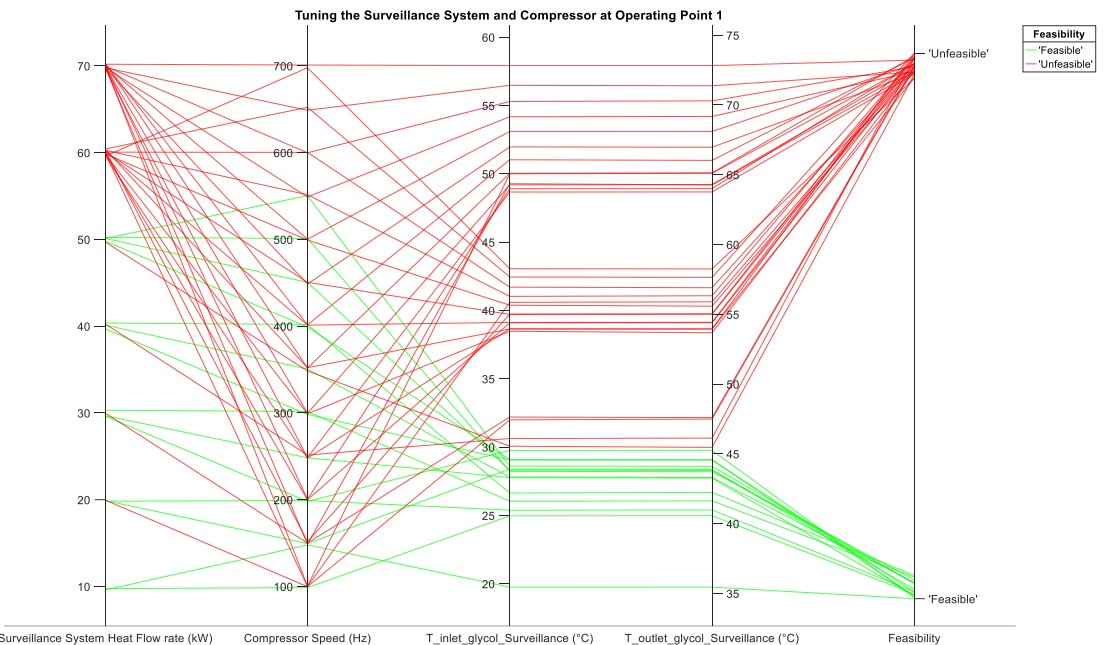

**Figure 7.** Cooling system parameters for tuning of the surveillance system heat flow rate, Q_(surveillance_sys) (kW) and compressor speed (Hz) at operating point 1. T_inlet_glycol_Surveillance (°C) and T_outlet_glycol_Surveillance (°C) represent the temperature of ethylene glycol at the inlet and outlet of the surveillance system, respectively. Feasibility represents whether input constraint 1 has been met or not.

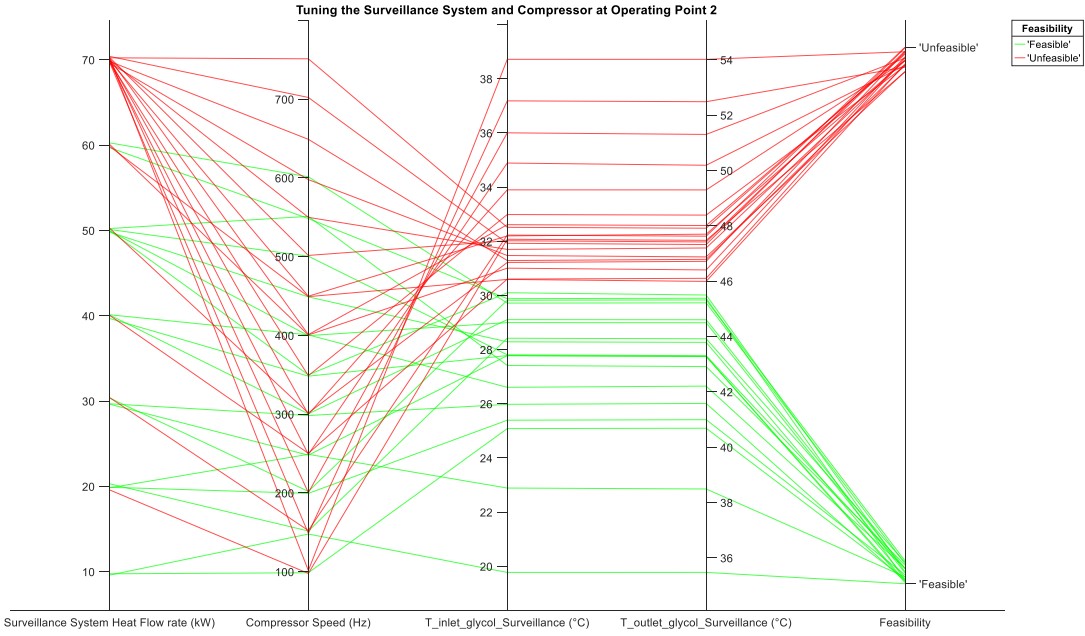

**Figure 8.** Cooling system parameters for tuning of the surveillance system heat flow rate, $\dot{Q}_{surveillance\_sys}$ (kW), and compressor speed (Hz) at operating point 2. T_inlet_glycol_Surveillance (°C) and T_outlet_glycol_Surveillance (°C) represent the temperature of ethylene glycol at the inlet and outlet of the surveillance system, respectively. Feasibility represents whether input constraint 1 has been met or not.

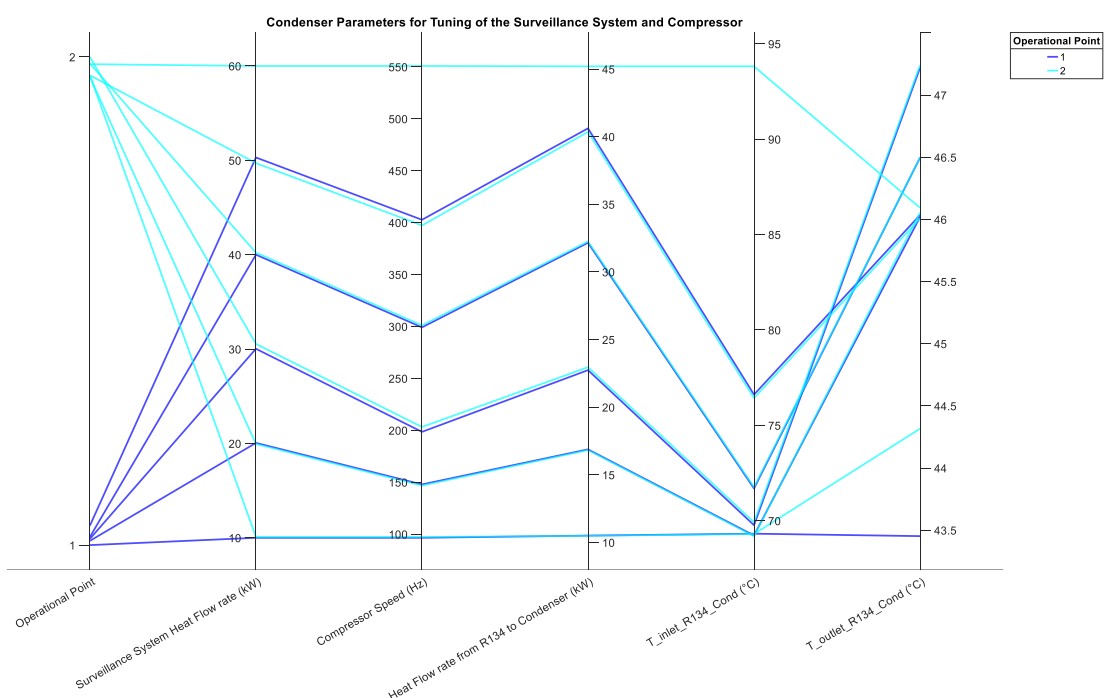

**Figure 9.** Condenser parameters for tuning of the surveillance system heat flow rate, $\dot{Q}_{surveillance\_sys}$ (kW) and compressor speed (Hz) at operating points 1 and 2. T_inlet_R134_Cond (°C) and T_outlet_R134_Cond (°C) represent the temperature of R134 at the inlet and outlet on the primary side of the condenser, respectively.

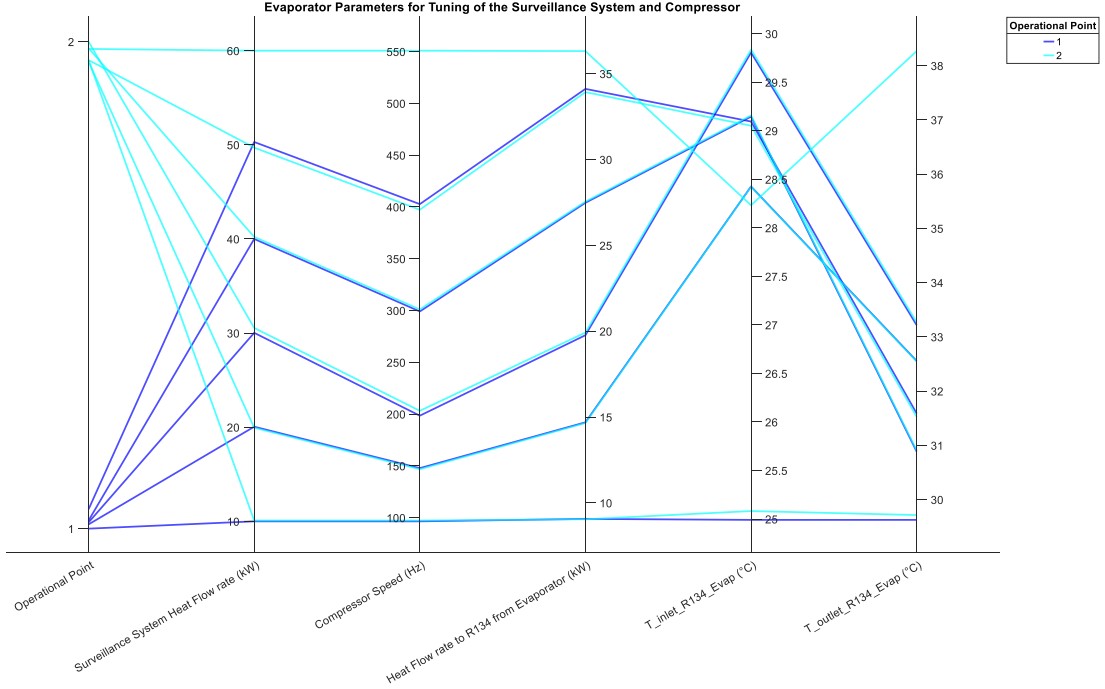

**Figure 10.** Evaporator parameters for tuning of the surveillance system heat flow rate, $\dot{Q}_{surveillance\_sys}$ (kW) and compressor speed (Hz) at operating points 1 and 2. T_inlet_R134_Evap (°C) and T_outlet_R134_Evap (°C) represent the temperature of R134 at the inlet and outlet on the primary side of the evaporator, respectively.

### 5.3. Performance of the Ram Air Heat Exchanger

Figure 11 displays the performance parameters for the ram air heat exchanger. For each $\dot{Q}_{surveillance\_sys}$, the results displayed in this figure are at the lowest compressor speed that does not violate input constraint 1. The compressor speed is obtained from Figures 7 and 8. The fourth axis in Figure 11 represents the heat flow rate from ethylene glycol to the ram air heat exchanger. In comparison to the condenser and evaporator, the ram heat exchanger has the highest heat flow rates. This is the heat load to be dispelled from the aircraft into the atmosphere. Therefore, these heat flow rates represent the cooling capacity of the system. At operating point 2 for $\dot{Q}_{surveillance\_sys}$ of 60 kW, the ram air heat exchanger must dump 69 kW overboard. This is an indication of the bottleneck of the cooling system at hand. The secondary side (ram air side) of the heat exchanger is the only terminal heat sink for the cooling system. The simulation results demonstrate that regardless of the performance of the refrigeration cycle (VCS), the true workhorse of the cooling system is the ram air heat exchanger. It is the deciding factor that determines if the system should be accepted or dismissed for the application at hand. The ram air heat exchanger could have been modelled by itself for a range of inlet temperatures above 57 °C for the ethylene glycol mixture. However, this would have only yielded the theoretical limits of the cooling system. Modelling the complete system and capturing the phase change of refrigerant R134 in the condenser and evaporator yields more realistic limits of the cooling capacity of the system.

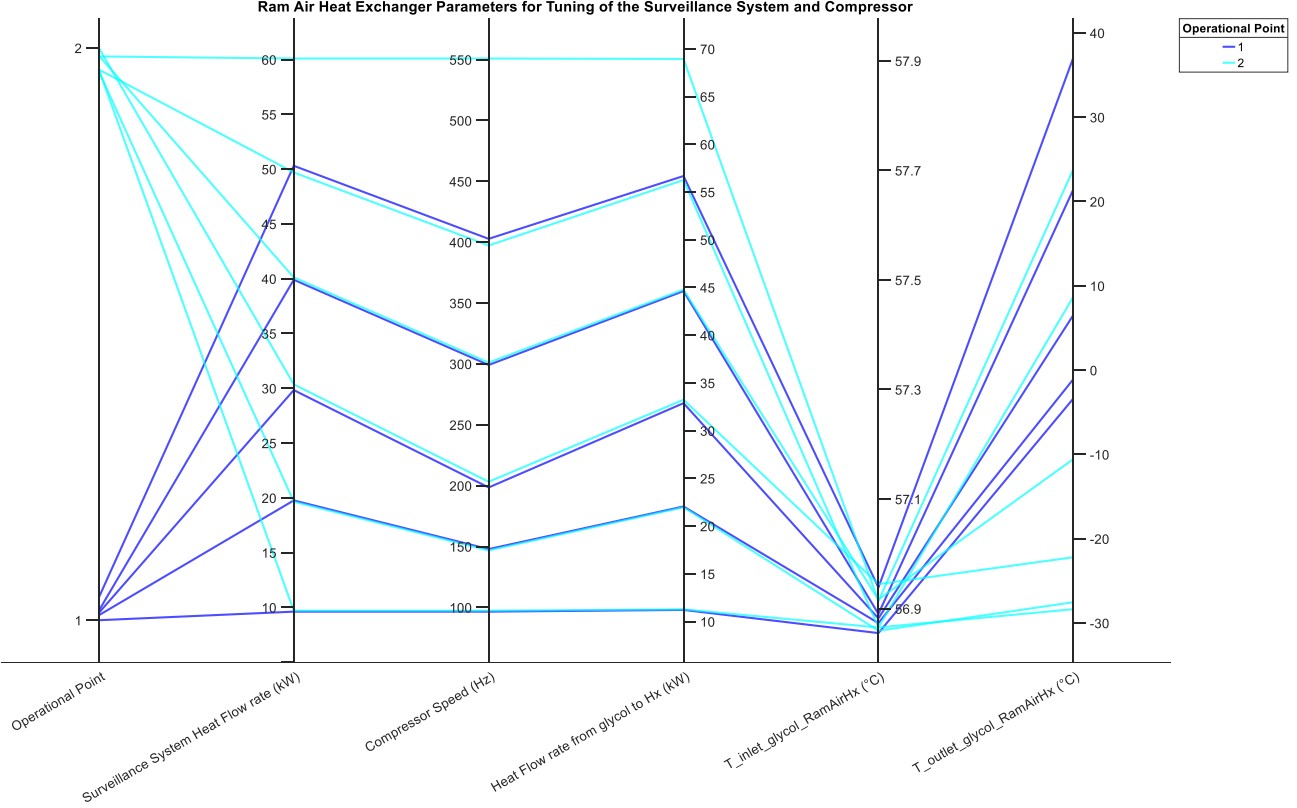

**Figure 11.** Ram air heat exchanger parameters for tuning of the surveillance system heat flow rate, $\dot{Q}_{surveillance\_sys}$ (kW) and compressor speed (Hz) at operating points 1 and 2. T_inlet_gylcol_RamAirHx (°C) and T_outlet_glycol_RamAirHx (°C) represent the temperature of ethylene glycol at the inlet and outlet on the primary side of the ram air heat exchanger, respectively.

In Figure 11, the fifth and sixth axes represent the temperature of ethylene glycol at the inlet, and outlet, respectively on the primary side of the ram air heat exchanger. The inlet temperature stays constant due to the bypass valve adjusting the mass flow rate (and thereby the pressure) of the ethylene glycol mass entering the ram air heat

exchanger. However, for the same $\dot{Q}_{surveillance\_sys}$, the outlet temperature differs from 25 °C to 32 °C between operating points 1 and 2. This would be expected since the inlet ram air temperature differs by 25 °C between operating points 1 and 2 as noted from Table 1. Therefore, the ethylene glycol is cooled to a lower temperature at operating point 2 for all $\dot{Q}_{surveillance\_sys}$.

At each $\dot{Q}_{surveillance\_sys}$, running the compressor at the lowest possible speed while not violating input constraint 1 reduces the power output (and thereby heat flow rate) from the compressor. To quantify this heat load, at operating point 1 for a $\dot{Q}_{surveillance\_sys}$ of 40 kW, the compressor speed was increased in increments of 10 Hz ranging from 300 Hz to 400 Hz. The performance parameters of the ram air heat exchanger for this tuning range are shown in Figure 12. The second, third, and fourth axes of Figure 12 represent the same parameters of the ram air heat exchanger as do the fourth, fifth, and sixth axes, respectively, in Figure 11. From Figure 12, it can be noted that for a 100 Hz increase in compressor speed, there is a 1.67 kW increase in heat flow rate from ethylene glycol. Therefore, for this specific simulation run, a 1 Hz increase in compressor speed resulted in a 16.7 W increase in compressor heat flow rate.

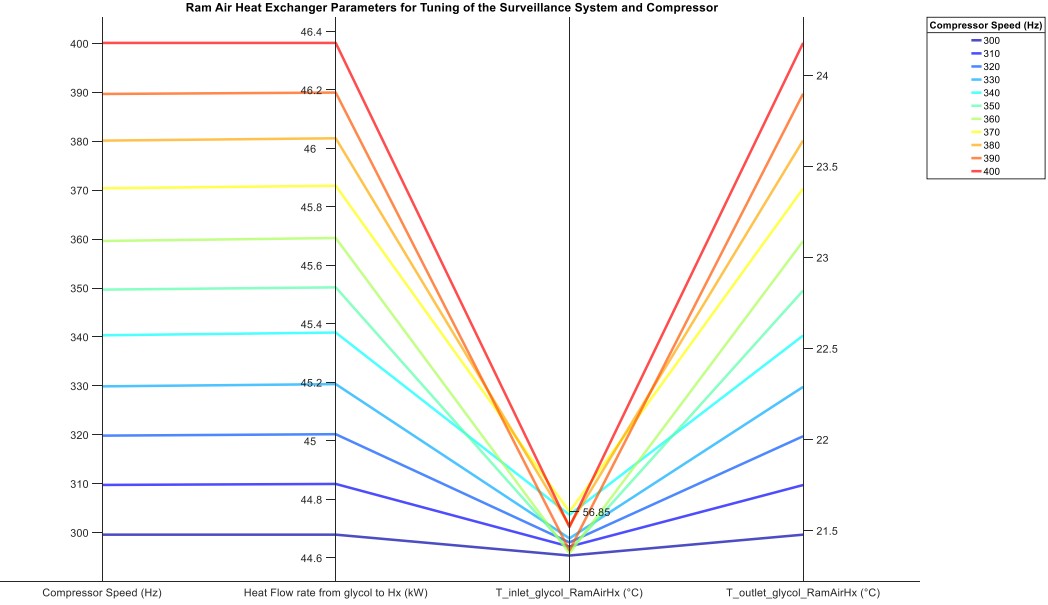

**Figure 12.** Ram air heat exchanger parameters for tuning of the compressor speed (Hz) at operating point 1 for a surveillance system heat flow rate, $\dot{Q}_{surveillance\_sys}$ of 40 kW. T_inlet_gylcol_RamAirHx (°C) and T_outlet_glycol_RamAirHx (°C) represent the temperature of ethylene glycol at the inlet and outlet of the primary side of the ram air heat exchanger, respectively.

### 5.4. Performance of the Vapor Cycle System

The performance of the VCS for its four main processes—evaporation, compression, condensation, and expansion—are shown in the pressure–enthalpy (p-h) diagrams of Figures 13 and 14 for operating points 1 and 2, respectively. For each $\dot{Q}_{surveillance\_sys}$, the results displayed in both figures are at the lowest compressor speed that does not violate input constraint 1. For all simulations cases, it can be noted that R134 leaves the evaporator and enters the compressor as a superheated vapor. This is because point 1 in all p-h diagrams are on the superheated vapor side of the p-h curve of R134. Similarly, R134 leaves the condenser as a sub-cooled liquid in all simulation cases since point 3 is on the liquid side of the p-h curve of R134. Heat rejection from R134 in the condenser (point 2 → point 3) and heat absorption by R134 in the evaporator (point 4 → point 1) do not occur at constant pressure for any simulation case. This becomes more apparent at higher $\dot{Q}_{surveillance\_sys}$. At operating point 2 for a $\dot{Q}_{surveillance\_sys}$ of 50 kW, a 71 kPa pressure drop is noted from point

2 to point 3 and 110 kPa pressure drop from point 4 to point 1. While at operating point 1 along with pressure drops, the superheat temperature of R134 increases from 30.9 °C for a $\dot{Q}_{surveillance\_sys}$ of 40 kW to 40.7 °C for a $\dot{Q}_{surveillance\_sys}$ of 50 kW. Therefore, for increasing $\dot{Q}_{surveillance\_sys}$, the VCS behaves less like an ideal refrigeration cycle.

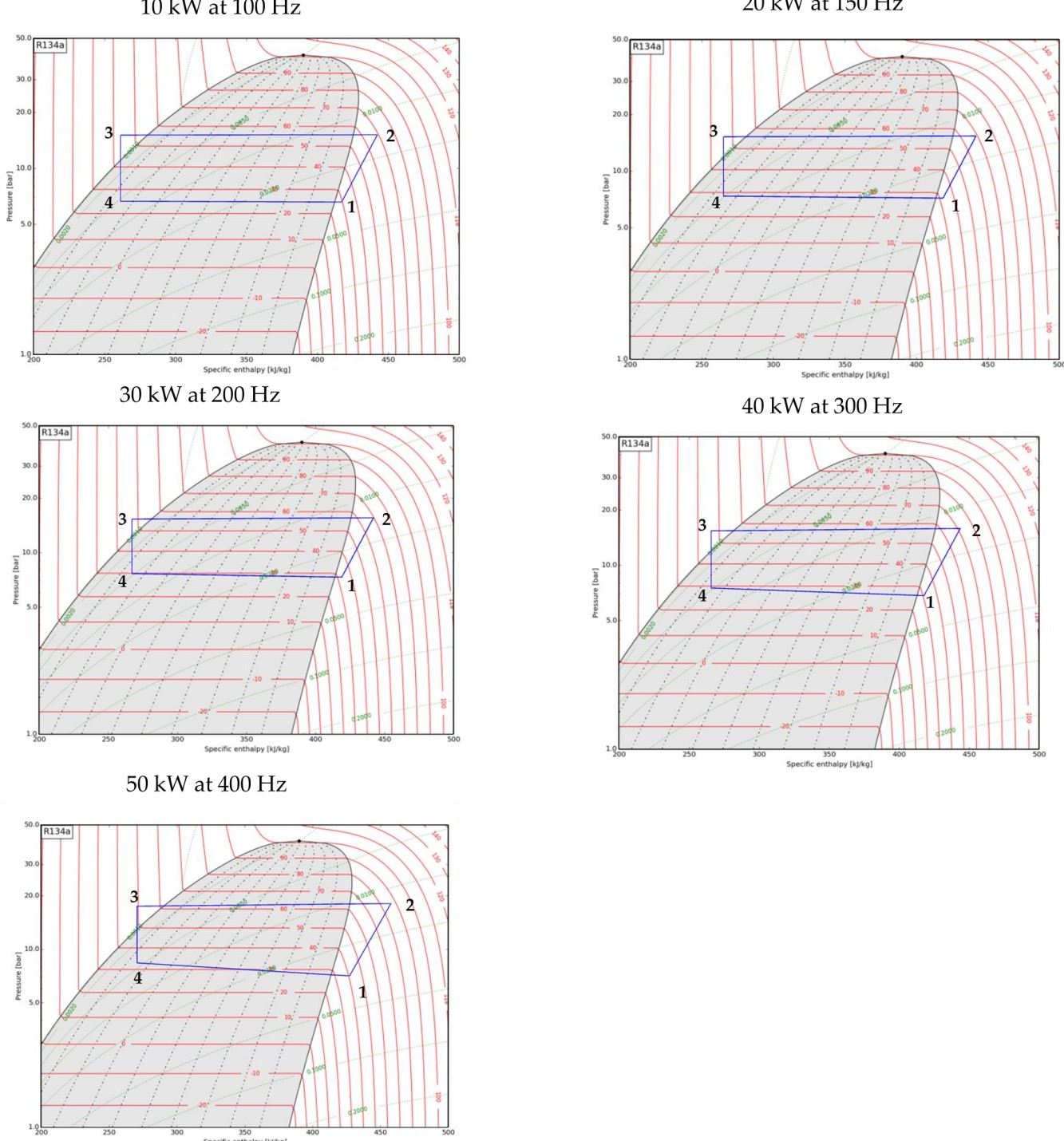

**Figure 13.** Pressure–enthalpy diagrams for the VCS at operating point 1 for surveillance heat flow rates, $\dot{Q}_{surveillance\_sys}$ of 10 kW, 20 kW, 30 kW, 40 kW, and 50 kW. Point 1: evaporator outlet, point 2: compressor outlet, point 3: condenser outlet, and point 4: thermostatic valve outlet.

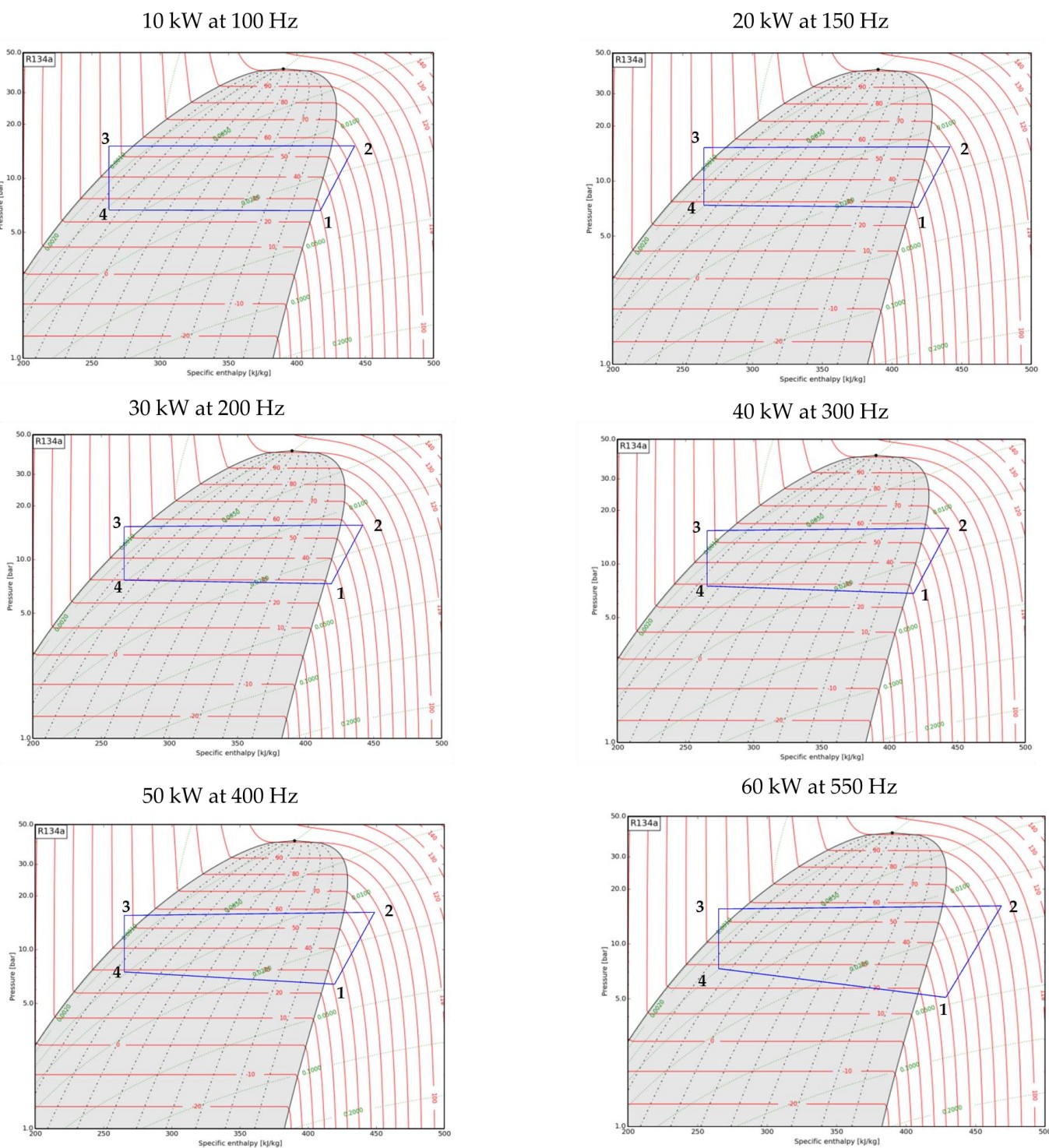

**Figure 14.** Pressure–enthalpy diagrams for the VCS at operating point 2 for surveillance heat flow rates of 10 kW, 20 kW, 30 kW, 40 kW, 50 kW, and 60 kW. Point 1: evaporator outlet, point 2: compressor outlet, point 3: condenser outlet, and point 4: thermostatic valve outlet.

## 6. Concluding Remarks

A two-part method to evaluate an aircraft cooling system consisting of a VCS was demonstrated in this paper. The first part focused on a parameter tuning study set-up and demonstrated how after identifying the operating conditions, constraints, and requirements, the only cooling system parameter available for tuning was the VCS compressor speed. The second part focused on a modelling and solving strategy for the cooling system and showed

how the capacity of an aircraft cooling system was impacted by tuning the VCS compressor speed (Hz) for a surveillance system heat flow rate from 10 kW to 70 kW. The tuning resulted in the maximum cooling capacity of the system being determined at both operating points. Along with the overall performance of the system, the performance of components is also shown in this paper; the performance of the heat exchangers of the cooling system and the VCS are shown. Further, this paper shows how the ram heat exchanger is the true workhorse of the cooling system. However, if only the ram air heat exchanger was modelled instead of the complete cooling system, only the theoretical cooling limit of the system would have been obtained. By modelling the complete system and choosing the appropriate level of detail for the system components, a more realistic limit of the cooling capacity was obtained. Finally, this paper demonstrates that for each $\dot{Q}_{\text{surveillance\_sys}}$ by running the compressor at the lowest possible speed while not violating input constraint 1 reduces the power output (and thereby heat flow rate) from the compressor. Therefore, the results from this two-part method can be used to design a control strategy for the compressor of the cooling system. In a larger context, the two-part method and the results analysis presented in this study can serve as a preliminary method for aircraft VCS control optimization studies.

**Author Contributions:** Conceptualization, A.D.D. and I.S.; methodology, A.D.D. and D.A.; software, A.D.D. and D.A.; validation, A.D.D., D.A. and I.S.; formal analysis, A.D.D.; investigation, A.D.D.; resources, A.D.D.; data curation, A.D.D.; writing—original draft preparation, A.D.D.; writing—review and editing, A.D.D., D.A. and I.S.; visualization, A.D.D.; supervision, I.S.; project administration, A.D.D.; funding acquisition, A.D.D. All authors have read and agreed to the published version of the manuscript.

**Funding:** This research was funded by VINNOVA (The Swedish Agency for Innovation Systems) under grant number 2019-02761.

**Data Availability Statement:** Data are contained within the article.

**Acknowledgments:** The authors would like to extend their gratitude to Ingela Lind, Hampus Gavel, and Michael Säterskog from Saab AB, Sweden. Ingela Lind contributed her expert knowledge on system modelling and simulation that was highly valuable to the methodology and results analysis for this paper. Michael Säterskog supported the formulation and definition of input constraints for this study. Hampus Gavel reviewed the paper and provided valuable feedback.

**Conflicts of Interest:** Adelia Darlene Drego was employed by Saab AB. Daniel Andersson was employed by Modelon AB. The remaining author declares that the research was conducted in the absence of any commercial or financial relationships that could be construed as a potential conflict of interest. The authors declare that this study was funded by VINNOVA. The funders had no role in the design of the study; in the collection, analyses, or interpretation of data; in the writing of the manuscript; or in the decision to publish the results.

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
