# Peer review of "Parameter Tuning of a Vapor Cycle System for a Surveillance Aircraft"

_aerospace, doi:10.3390/aerospace11010066_

Round 1
Reviewer 1 Report
Comments and Suggestions for Authors
The paper describes the process of analysing and predicting the behaviour of a Vapor Cycle System applied to the cooling of surveillance aircraft equipment (with potential retrofitting), by means of system modelling and simulation. Particularly, the focus on the impact of constraints on system tuning is highlighted, as well as the importance of a model-based approach towards the analysis of system cycle.
The paper is in line with Journal scope and is relevant for publication, since it can be interesting for the scientific community.
However, in the present form, the manuscript is highly focused on a specific application of the method, while the methodological background, which should be the core for every journal publication, seems to be poor or lacking. This Reviewer suggests to improve the description of the methodology adopted to carry out the exercise on the case study, highlighting the novelty of the approach, before moving to the description of the cooling system under design (i.e. before current section 2). Elements such as methodology workflow, main steps of the approach, management of requirements and constraints in the digital environment, as well as problem formulation should be described before moving to the example of application.
Also, some parts of the paper may need further re-organisation. In particular, Section 1 shall state more clearly which is the structure of the paper (Section 1.2 is not very precise). Section 3 to 7 contains a lot of sub-paragraphs which sometimes are short or redundant. For example, Sections 3 and 4 can be merged, and the same applies to Sections 7 and 8. It is also not really easy to follow the description of some results in the text, such as those described in Section 8.1 and referred to Figures 6-7.
Conclusions shall refer more to the method, highlighting strong points and potential bottlenecks of the approach, and not only to the specific case study, as discussed before.
Comments on the Quality of English LanguageQuality of English is fair, even if a moderate overview of the style may be an advantage.
Author Response
Please find the attached pdf document with the response to Reviewer 1's comments.

Reviewer 2 Report
Comments and Suggestions for Authors
Review on the manuscript “Parameter Tuning of a Vapor Cycle System for a Surveillance Aircraft”, by Drego et al., ID: aerospace-2686714
In this work vapor cycle system for surveillance aircraft was numerically simulated and studied for two different aircraft operating conditions in terms of altitude and Mach number. The topic is worth of investigation, the article is well written and well organized. It contains several results and an in-depth analysis which is well presented and discussed. However, there are some minor concerns that authors should consider/clarify before publication:
1- First paragraph of introduction provides very interesting information however does not contain any citation to validate that content. References should be used to support the first paragraph of the introduction.
2- In 2nd page, 2nd paragraph, 12th line, authors start a sentence with: “[18] (pp. 4-6, 12) and [19](p.800) categorize…” When authors mention a work like this, they should give proper credit to the authors of that work and should mention the name of the authors. Therefore, this sentence should be corrected to: Van Heerden et al. [18] (pp. 4-6, 12) and Pal & Severson [19] (p.800) categorize…”
3- The cooling system studied in the current work is composed by a liquid loop system with ethylene glycol and a vapor cycle system with R134. Please justify and explain the choice of these coolants.
4- Authors performed the study for two operating points in which the first simulates an altitude of 6.5 km and a flight Mach number of 0.4 and the second simulates an altitude of 11 km and a flight Mach number of 0.55. Please explain why you defined these conditions.
5- Authors should compare their simulated results with experimental results from the literature in order to validate the proposed cooling system. Which differences will we observe from the simulated system to a real implementation? Could you estimate an uncertainty or error quantity?
6- Authors present several results and explain them, but they do not compare them with literature. Please compare your results with other published works, and emphasize the advantages of your system against the reported ones.
7- Figures 12 and 130 present very low quality and the numbers and legend of the axis are not visible. Please improve the quality of these images.
Comments on the Quality of English Language
The article is well written just some minor mistakes are found. The article should be proofread.
Author Response
Please find attached a pdf document with the response to Reviewer 2's comments.

Round 2
Reviewer 1 Report
Comments and Suggestions for Authors
The Reviewer would like to thank the Authors for their effort in updating the content of the manuscript following the provided suggestions. The work is now organized in a better way and the focus on the method is appreciated. The paper is now ready for publication.
Comments on the Quality of English LanguageQuality of English is fair
Reviewer 2 Report
Comments and Suggestions for Authors
Authors addressed all my coments and improved the manuscript, in my opinion it can be now accepted for publication.